# "Energetics of the outer retina II: Calculation of a spatio-temporal energy budget in retinal pigment epithelium and photoreceptor cells based on quantification of cellular processes"

Christina Kiel[1]*, Stella Prins[2], Alexander J. E. Foss[3], Philip J. Luthert[2,4]*

1 Department of Molecular Medicine, University of Pavia, Pavia, Italy, 2 UCL Institute of Ophthalmology, University College London, London, United Kingdom, 3 Department of Ophthalmology, Nottingham University Hospitals NHS Trust, Nottingham, United Kingdom, 4 NIHR Moorfields Biomedical Research Centre, University College London, London, United Kingdom

* christina.kiel@unipv.it (CK); p.luthert@ucl.ac.uk (PJL)

**Data Availability Statement:** All relevant data are within the manuscript and its Supporting information files.

## Abstract

The outer retina (OR) is highly energy demanding. Impaired energy metabolism combined with high demands are expected to cause energy insufficiencies that make the OR susceptible to complex blinding diseases such as age-related macular degeneration (AMD). Here, anatomical, physiological and quantitative molecular data were used to calculate the ATP expenditure of the main energy-consuming processes in three cell types of the OR for the night and two different periods during the day. The predicted energy demands in a rod dominated (perifovea) area are $1.69 \times 10^{13}$ ATP/s/mm$^2$ tissue in the night and $6.53 \times 10^{12}$ ATP/s/mm$^2$ tissue during the day with indoor light conditions. For a cone-dominated foveal area the predicted energy demands are $6.41 \times 10^{12}$ ATP/s/mm$^2$ tissue in the night and $6.75 \times 10^{12}$ ATP/s/mm$^2$ tissue with indoor light conditions during daytime. We propose the likely need for diurnal/circadian shifts in energy demands to efficiently stagger all energy consuming processes. Our data provide insights into vulnerabilities in the aging OR and suggest that diurnal constraints may be important when considering therapeutic interventions to optimize metabolism.

## Introduction

Cells transform energy and use it to drive cellular processes that essentially keep living systems in a far-from-equilibrium steady-state. While the systematic mapping of protein interactions ('network biology') and the knowledge about cellular biochemistry and cell biology is ever increasing, little is known about the energetic costs associated with specific cellular processes. A critical step in achieving this is to move from a purely qualitative to a quantitative and integrative ('systems biology') framework.

The main role of the OR is phototransduction and this depends upon complex metabolic exchanges between photoreceptor (PR) inner and outer segments (OS), the retinal pigment

**Funding:** This work was supported by a Moorfields Eye Charity (Grant GR001345).

**Competing interests:** The authors have declared that no competing interests exist.

epithelium (RPE) and the choriocapillaris of the choroidal circulation. The maintenance of ion gradients, particularly the so called dark current for PRs, contributes substantially to the OR's energy needs [1, 2]. There are, however, many cell-general processes, such as turnover of biomass (e.g. transcription, ribosome production, translation) and organelles, cytoskeletal dynamics and transport whose energy requirements have not been systematically mapped out in the major cell types of the OR. Here, we use quantitative molecular (e.g. gene expression), anatomical (e.g. compartment volumes), and biochemical (e.g. catalytic rate constants) data to generate the first near complete energy budget for three critical OR cell types. We used estimates derived from the literature and published databases to deduce the energy demands of the rod and cone PRs and of the RPE. We propose that as well as the dark current, some of the other energy demanding processes, occur at only a certain part of the 24 hours cycle. Accordingly, we not only list the energy demanding processes and the amount of energy consumed but also estimate which part of the 24 hours cycle each process occupies.

## Methods

### Cell volume of photoreceptor cells

Volumes of rod PR cells were estimated using anatomical data available in the literature. The approximate length of rPR cells is 100 µm and the width is 2 µm (BNID 108246, 109684), which includes the axon (which we assumed to be 10 µm). Excluding the axon region, results in a subvolume of 283 µm$^3$. The nuclear volume was estimated based on EM pictures (80 µm$^3$) [3]. The axon volume was calculated based on a cylinder with length of 10 µm and a diameter of 0.45 µm (= ~ 1.59 µm$^3$) [4]. The synaptic terminal volume was calculated based on a sphere with diameter of 4 µm (33,51 µm$^3$). The sum of the volumes of these four anatomical compartments results in a total intracellular volume of ~400 µm$^3$. Cone photoreceptors vary in size greatly from the fovea to the periphery [5, 6]. As we were particularly interested in establishing the most highly constrained energy demands, we have carried out our quantitative analyses on foveal cones which are packed at a very high density. Foveal cones appear similar to rods and for simplicity we have assumed the same total volume for a foveal cone and a more peripheral rod.

### Biomass composition of cells

To obtain estimates for the main cellular biomolecules in PR and RPE cells in terms of weight (in gram) and number of molecules per cell, we used the biomass compositions obtained in mammalian cells (e.g. Hela, CHO) and assumed that similar percentages apply to PR and RPE cells (S1 Table). For Hela cells, we obtained a ratio of mass (= $2.3 \times 10^{-9}$ g; Bionumbers ID 103720) to volume (= 3000 µm$^3$; Bionumbers ID 103725) of $7.67 \times 10^{-13}$ g/µm$^3$. Applying this ratio to the volume of PR and RPE cells, results in a cell mass of $3.32 \times 10^{-10}$ g (PR) and $1.95 \times 10^{-9}$ g RPE). Of the total cell weight, weight percentages (based on CHO cells): water (70%), protein (15%), lipid (5.4%), RNA (2.4%), DNA (0.6%), carbohydrates (0.3%), inorganic ions (1%), and other biomolecules (2.3%) (Bionumbers ID 107131) were used [7]. Lipids were subclassified into main cellular phospholipids and cholesterol [7]. RNA was subclassified into main RNA classes for a typical mammalian cell [8].

### Prediction of protein expression levels for PR and RPE cells

Protein expression levels were estimated from gene expression. Human gene expression data were obtained from single cell RNA sequencing (scRNAseq) for RPE cells [9]. Gene names by were matched to 19300 protein coding genes [10, 11]. Likewise, gene expression levels from

scRNAseq data for rod and cone PR cells were obtained [12]. For each dataset, expression levels were summed to calculate an expression level in % for each of the 19300 genes.

To calculate the number of proteins (and volumes occupied) at each subcellular localization, we used the 19 protein-specific subcellular localizations (groups A to S) as defined in the EnerSysGO database for 19300 protein-coding genes [11]. The expression values for each of the 19300 genes in RPE cells in % were summed for all genes in the "Extracellular" localization (group A) and for all "Remaining" localizations (groups B to S). This resulted in 16.5% extracellular proteins and 83.5% remaining proteins (for RPE cells). As $7.45 \times 10^9$ proteins correspond to the 83.5% remaining proteins, an additional $7.45 \times 10^9 \times (83.5/16.5 = 5.1) = 1.48 \times 10^9$ proteins was assigned to the extracellular proteins. This results in $8.96 \times 10^9$ proteins in the RPE cell including the extracellular space. Similar calculations were performed for rPR and cPR cells. Using the total number of proteins per cell in the intracellular volume (all genes in groups B to S) the number of copies for each protein was calculated based on the gene expression levels (in %). Protein concentrations (in mol $L^{-1}$) were calculated based on cellular volumes (see S2 Table).

## Anatomy of two "typical" 1 mm$^2$ patches of the retina

There is significant variation of RPE, rod and cone density from the fovea to the periphery. The average human RPE cell has an area of 221 μm$^2$ (= 4525 cells/mm$^2$) [13]. The RPE cell count per unit area varies from 4220 cells/mm$^2$ at the fovea to 1600 in the far periphery [14]. Rod peak density is close 150,000 cells/mm$^2$ (at 15o to 20o eccentricity) and falling to half that at 70,000 cells/mm$^2$ in the periphery [15, 16]. Cone density peaks at the fovea at 199,000/mm$^2$ but is highly variable (100,000–324,00 cones/mm$^2$) [17] and a separate estimate provides a figure of 150,000 [16]. These figures are age dependent with an estimated annual loss of photoreceptor loss of 0.2–0.4% per year [18]. Hence, a typical patch of central retina has ~4000 RPE cells and 150,000 PR per mm$^2$. For the energy budget, we provide calculations at two "typical" retinal patches: one rod-dominated 1 mm$^2$ patch with 4000 RPE cells and 150000 rPR cells, and one cone dominated 1 mm$^2$ patch with 4000 RPE cells, 80000 cPR cells, and a small number of rPR cells being present (1000). Based on the assumption that protein content scales with cell volume (taking into account weighting for different relative volumes of subcellular compartments) the energy budget can be recalculated for different cell size and cell density at different eccentricities.

## Error and uncertainty estimation associated with ATP fluxes

There are different sources of errors and uncertainties linked to the final estimates of ATP demand fluxes. When ATP flux calculations require enzyme abundancies, there are uncertainties linked to those values as they are estimated from mRNA expression abundancies. The correlation between mRNA and protein abundance is weak [19] and there might be individual proteins that have significantly higher or lower protein levels as estimated from mRNA. When using catalytic activities for calculating ATP fluxes, there are errors linked to the measurement of $k_{cat}$ (e.g. enzyme impurities; generally < 10% or the use of non-physiological conditions for the assay). When ATP fluxes are calculated based on a specific demand for a process (e.g. protein or lipid turnover) there are errors associated with determination of demand and these measurements are often technically challenging. Moreover, sometimes those data are not available for the cell types analysed in this work and estimates had to be taken from other tissues (e.g. liver). It is not possible to quantify an error linked to those data. Finally, there are uncertainties linked to many assumptions we take for processes where the dynamics is less studied. This applies in particular to processes linked to cytoskeletal dynamics, where we not only had

## Results

### Quantitative data required for calculating the energy budget

It is often not possible to get direct measurement of energy requirements for a specific process [20]. In these cases, we obtained estimates based on measurements of demand. Examples would include the turnover of cell biomass components such as proteins and lipids. A second approach used was to take the product of relevant total enzyme abundances ($E_t$) and catalytic constants ($k_{cat}$) catalytic constant) to estimate maximum reaction rates ($V_{max}$), with $V_{max} = k_{cat} * E_t$. $k_{cat}$ is also known as the turnover number and is the maximum number of substrate molecules turned into product per unit time per active site of an enzyme (or transporter). $V_{max}$ provides an upper limit to the rate for a given enzyme abundance. Hence, in those cases, the ATP fluxes would also present upper limits.

Both approaches require quantitative molecular data (e.g. protein abundances; see methods; S2 Table) and to generate a budget per cell, the volumes of the cell types considered are also required. A total intracellular volume of ~400 mm$^3$ was calculated from anatomical estimates for a typical rPR cell. Most cone PR cells have short outer segments but have larger inner segments, but we were particularly interested in the long, thin foveal cones [21] and to simplify, a similar volume of was therefore assigned to cPR cells. The total intracellular volume of RPE cells was obtained from Keeling et al [22], where the average volume of five RPE cells was estimated from serial block face scanning electron microscopy images (2548 μm$^3$).

To obtain the number of proteins that occupy specific cellular volumes, we used the following relation between volume occupied and the number of proteins [23]: 3 x 10$^6$ = (proteins/μm$^3$).

This results in 1.21 x 10$^9$ intracellular proteins (for rPR and cPR cells) and 7.64 x 10$^9$ (for one RPE cell). These numbers agree well with estimates obtained from cellular biomass composition and an average weight of one protein of 6.39 x 10$^{-20}$ g, which results in 7.79 x 10$^8$ proteins (for rPR and cPR) and 4.59 x 10$^9$ proteins (for RPE) (S1 Table).

ATP hydrolysis to ADP was used as the main energy currency. Where other currencies or intermediates with high chemical potential energy (e.g. GTP, NAD+, Acetyl-CoA) were needed in cellular processes, 'currency conversion rates' were applied (S3 Table).

### Cellular processes and their diurnal variation

Many cellular processes follow circadian or diurnal variations [24]. Indeed, this has been proposed to promote metabolic efficiency [25]. The outer retina (OR) of the eye is unusual because of the high energy demands of photoreceptors in the dark (12-hour duration). We hypothesize that replacing biomass and 'maintenance' processes are done at other times when energy demands are less. We assumed that housekeeping activities such as protein turnover occur mainly during the day when overall energy demand is low. Turnover requires recycling of, for example, amino acids and so catabolic processes are assumed to precede anabolic ones. This fits with what is known with RPE cells, undertaking phagocytosis of the OS tips at dawn which are then broken down in lysosomes. This catabolic phase is presumed to be followed by an anabolic phase (i.e. protein synthesis, organelle biosynthesis) in the afternoon (6-hour duration), which would enable the recycling of components (e.g. amino acids). Table 1 summarizes cellular processes we considered for the energy budget and their proposed time frame.

**Table 1. Processes considered for the energy budget with subcellular localisation and proposed timing.**

| Process | Sub-process | Compartment | Circadian/ diurnal variation | | |
|---|---|---|---|---|---|
| | | | rPR | cPR | RPE |
| Plasma membrane ion transport | Na+/K+ ATPase | Plasma membrane | Fast NIGHT; basal DAY 1st 6 h & DAY 2nd 6 h | Fast NIGHT; basal DAY 1st 6 h & DAY 2nd 6 h | Constant activity NIGHT & DAY 1st 6 h & DAY 2nd 6 h |
| | Ca2+ ATPase | Plasma membrane | Fast NIGHT; basal DAY 1st 6 h & DAY 2nd 6 h | Fast NIGHT; fast DAY 1st 6 h & DAY 2nd 6 h | Slow NIGHT; fast DAY 1st 6 h; fast DAY 2nd 6 h |
| Cytoskeleton | Actin fiber polymerization and depolymerization | Cytosol | Slow NIGHT; slow DAY 1st 6 h; fast DAY 2nd 6 h | Slow NIGHT; slow DAY 1st 6 h; fast DAY 2nd 6 h | Slow NIGHT; fast DAY 1st 6 h; fast DAY 2nd 6 h |
| | Transport by myosin molecules on actin filaments | Cytosol | DAY 2nd 6 h | DAY 2nd 6 h | DAY 1st 6 h; DAY 2nd 6 h |
| | Tubulin fiber polymerization and depolymerization | Cytosol | DAY 2nd 6 h | DAY 2nd 6 h | DAY 1st 6 h; DAY 2nd 6 h |
| | Transport by kinesin molecules on microtubules | Cytosol | Low density, low k_cat NIGHT & DAY 1st 6 h; Intermediate density, average k_cat DAY 2nd 6 h | Low density, low k_cat NIGHT & DAY 1st 6 h; Intermediate density, average k_cat DAY 2nd 6 h | Low density, low k_cat NIGHT; Intermediate density, average k_cat DAY 1st 6 h & DAY 2nd 6 h |
| | Transport by dynein molecules on microtubules | Cytosol | Low density, NIGHT & DAY 1st 6 h; Intermediate density, DAY 2nd 6 h | Low density, NIGHT & DAY 1st 6 h; Intermediate density, DAY 2nd 6 h | Low density, NIGHT; Intermediate density, DAY 1st 6 h & DAY 2nd 6 h |
| | Membrane trafficking accounting for Rab and Arf GTPases | Cytosol | Half of proteins cycling, NIGHT & DAY 1st 6 h; All proteins cycling, DAY 2nd 6 h | Half of proteins cycling, NIGHT & DAY 1st 6 h; All proteins cycling, DAY 2nd 6 h | Half of proteins cycling, NIGHT; All proteins cycling, DAY 1st 6 h & DAY 2nd 6 h |
| | Rho GTPases | Cytosol | Half of proteins cycling, NIGHT & DAY 1st 6 h; all proteins cycling, DAY 2nd 6 h | Half of proteins cycling, NIGHT & DAY 1st 6 h; all proteins cycling, DAY 2nd 6 h | Half of proteins cycling, NIGHT; all proteins cycling, DAY 1st 6 h & DAY 2nd 6 h |
| Protein turnover | Messenger (m)RNA turnover–primary transcript production | Nucleus | DAY 2nd 6 h | DAY 2nd 6 h | DAY 2nd 6 h |
| | Messenger (m)RNA turnover–splicing and nuclear export | Nucleus | DAY 2nd 6 h | DAY 2nd 6 h | DAY 2nd 6 h |
| | Ribosome turnover–transport of ribosomal protein subunits into nucleus | Nucleus | DAY 2nd 6 h | DAY 2nd 6 h | DAY 2nd 6 h |
| | Ribosome turnover–production of rRNA in nucleus | Nucleus | DAY 2nd 6 h | DAY 2nd 6 h | DAY 2nd 6 h |
| | Ribosome turnover–processing of small subunit | Nucleus | DAY 2nd 6 h | DAY 2nd 6 h | DAY 2nd 6 h |
| | Ribosome turnover–processing of large subunit–nuclear steps | Nucleus | DAY 2nd 6 h | DAY 2nd 6 h | DAY 2nd 6 h |
| | Ribosome turnover–processing of large subunit–cytoplasmic steps | Cytosol | DAY 2nd 6 h | DAY 2nd 6 h | DAY 2nd 6 h |
| | Translation | Cytosol | DAY 2nd 6 h | DAY 2nd 6 h | DAY 2nd 6 h |
| | Protein chaperone folding | Cytosol | DAY 2nd 6 h | DAY 2nd 6 h | DAY 2nd 6 h |
| | Protein posttranslational modifications (PTMs) | All subcellular localisations | NIGHT and DAY | NIGHT and DAY | NIGHT and DAY |
| | Protein degradation via proteasome | Cytosol | NIGHT; DAY 1st 6 h; DAY 2nd 6 h | NIGHT; DAY 1st 6 h; DAY 2nd 6 h | NIGHT; DAY 1st 6 h; DAY 2nd 6 h |
| | Protein degradation via lysosome | Cytosol | DAY 1st 6 h | DAY 1st 6 h | DAY 1st 6 h |

(*Continued*)

**Table 1.** (Continued)

| Process | Sub-process | Compartment | Circadian/ diurnal variation | | |
|---|---|---|---|---|---|
| | | | rPR | cPR | RPE |
| Lipid turnover | Cholesterol production | Cytosol | DAY 2$^{nd}$ 6 h | Not considered | Not considered |
| | Phospholipid breakdown | Cytosol | DAY 1$^{st}$ 6 h | DAY 1$^{st}$ 6 h | DAY 1$^{st}$ 6 h |
| | Phospholipid production | Cytosol | DAY 2$^{nd}$ 6 h | DAY 2$^{nd}$ 6 h | DAY 2$^{nd}$ 6 h |
| | Maintaining cell membrane asymmetry (flippases) | Plasma membrane | NIGHT and DAY | NIGHT and DAY | NIGHT and DAY |
| Intracellular ion transport | Lysosomes | Cytosol | DAY 1$^{st}$ 6 h | DAY 1$^{st}$ 6 h | DAY 1$^{st}$ 6 h |
| | Endoplasmic reticulum and golgi | Cytosol | DAY 2$^{nd}$ 6 h | DAY 2$^{nd}$ 6 h | DAY 2$^{nd}$ 6 h |
| DNA repair | All repair mechanisms | Nucleus | DAY 2$^{nd}$ 6 h | DAY 2$^{nd}$ 6 h | DAY 2$^{nd}$ 6 h |
| Retina-specific processes | Visual transduction—Transducin activation | Rod outer segments | DAY 1$^{st}$ 6 h, DAY 2$^{nd}$ 6 h | Not considered | Never |
| | Visual transduction—PDEdelta activation and cGMP hydrolysis (GC) | Rod outer segments, Cone outer segments | DAY 1$^{st}$ 6 h; DAY 2$^{nd}$ 6 h | DAY 1$^{st}$ 6 h; DAY 2$^{nd}$ 6 h | Never |
| | Visual transduction—Phosphorylation of Rhodopsin by Rhodopsin kinase | Rod outer segments | DAY 1$^{st}$ 6 h; DAY 2$^{nd}$ 6 h | Not considered | Never |
| | Visual transduction—Retinal recycling | Rod outer segments; RPE cytosol | DAY 1$^{st}$ 6 h; DAY 2$^{nd}$ 6 h | Not considered | DAY 1$^{st}$ 6 h; DAY 2$^{nd}$ 6 h |
| | Synaptic transmission (dark) | Cytosol | NIGHT | NIGHT | Never |

### Constructing an energy budget for three retinal cell types

The detailed calculations and assumptions are summarized in S1 Text in S1 File, S4 Table and Table 1 and the main results are summarized here.

**Plasma membrane ion transport.** The sodium-potassium ATPase (Na$^+$/K$^+$-ATPase) in the plasma membrane (PM) pumps Na$^+$ out of the cell coupled to K$^+$ influx at the expense of ATP (Fig 1(a)). Already in 1962, Whittam noted in rabbit brain slices that inhibiting the Na$^+$/K$^+$-ATPase halved oxygen consumption [26]. In rPR cells, influx of Na$^+$ and Ca2+ (which is first removed by the Na+/Ca2+ exchanger; the incoming sodium being removed by the Na$^+$/K$^+$-ATPase) into cells is particularly high in the dark ("dark current"; 40 pA) [27], which requires $1.02 \times 10^8$ ATP molecules/s/rod to compensate (Fig 1(b)). Taking the protein abundances of the Na$^+$/K$^+$-ATPase and its k$_{cat}$ (45 s$^{-1}$; see S1 File) gives a maximum flux of $2.65 \times 10^7$ ATP/s/cell which is 3.8-fold lower (and likely within experimental error) and suggest that in the dark the enzyme is working at its upper limit. The ATP flux drops under light conditions to ~$10^7$ ATP/s/cell [2, 28].

The cone V$_{max}$ based on Na$^+$/K$^+$-ATPase abundance and k$_{cat}$ (see above) is $4.23 \times 10^7$ ATP/s/cell, which is close to the physiological estimates of $5.0 \times 10^7$ ATP/s/cell at night [2]. This drops to $2.5 \times 10^7$ ATP/s/cell for day with indoor light conditions (Fig 1(c)) [2].

For RPE cells, as expected given their size, a higher ATP flux compared to rod and cones is required to support Na+/K+ ATPase ($1.07 \times 10^8$ ATP/s/cell based on demand [29]), which is slightly lower compared to what can maximally be delivered based on Na$^+$/K$^+$-ATPase abundance and k$_{cat}$ (see above) ($2.05 \times 10^8$ ATP/s/cell). Hence, we worked with the flux calculated by demand (Fig 1(d)).

Both rods and cones have an inner segment inward calcium current in the dark, which is removed by Ca$^{2+}$-ATPases. For rods this is 4 pA [30] and for cones close to 50 pA [2]. This

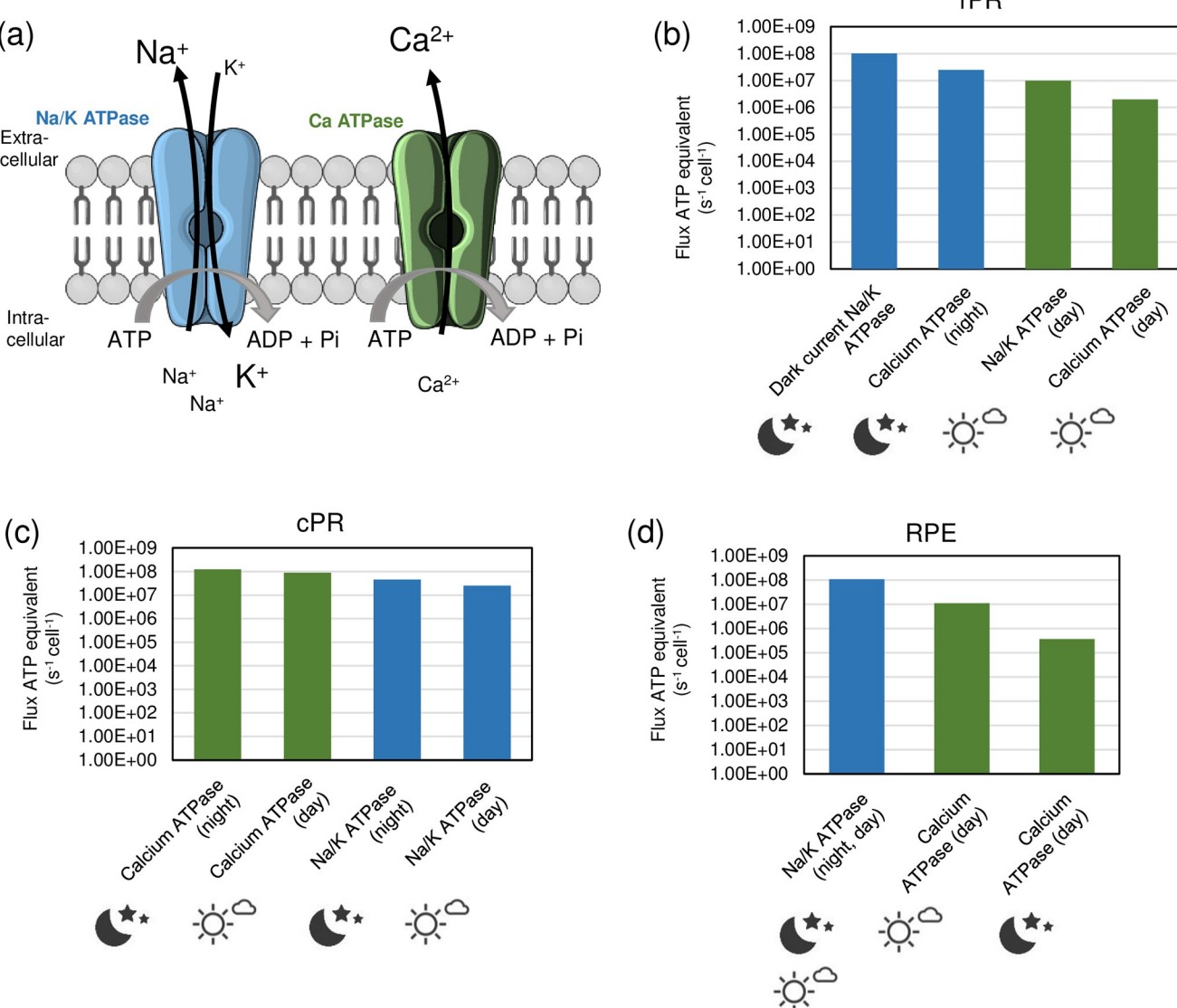

**Fig 1. Energy needed for PM ATPases.** (a) Schematic representation of the functioning of Sodium-potassium (Na/K) and Calcium (Ca) ATPases. The Na/K ATPase pumps for every ATP 3 Na$^+$ ions out and 2 K$^+$ ions in. The Ca ATPase pumps for every ATP one Ca$^{2+}$ ion out. (b-d) ATP fluxes per cell related to maintenance of PM ion gradients for rPR, cPR, and RPE at night and day. The Figure was partly generated using Servier Medical Art, provided by Servier, licensed under a Creative Commons Attribution 3.0 unported license.

results in a flux of $2.5 \times 10^7$ ATP/s/rod at night and of $2.0 \times 10^6$ ATP/s/rod in indoor light conditions [2], based on demand. In cones, the respective fluxes are more similar in the night ($1.25 \times 10^8$ ATP/s/cone) and in indoor conditions ($9 \times 10^7$ ATP/s/cone). In RPE cells, based on Ca$^{2+}$-ATPase abundance and enzyme activity that depends on Ca$^{2+}$ concentration, we obtained ATP demands in the range of $3.73 \times 10^5$ to $1.12 \times 10^7$ ATP/s/cell. Our hypothesis is that the flux in RPE cells is low in the dark and high in light because of voltage-gated channels that open with light [31].

Altogether, the dark current dominates the energy budget in rPR cells (Fig 1). In cPR cells Ca$^{2+}$-ATPase dominate and their activity is high also during daytime. In the RPE, Na$^+$/K$^+$-ATPases dominate the energy budget both in night and day.

**Cytoskeleton.** Actin cytoskeleton remodeling is critical for many cellular processes (including metabolism; Fig 2) [32]. In platelets, the cytoskeleton was demonstrated to be a major energy consumer, where 50% of ATP consumption was used to maintain the actin cytoskeleton [33]. It is also a major energy consumer in neurons [34]. To calculate the energy

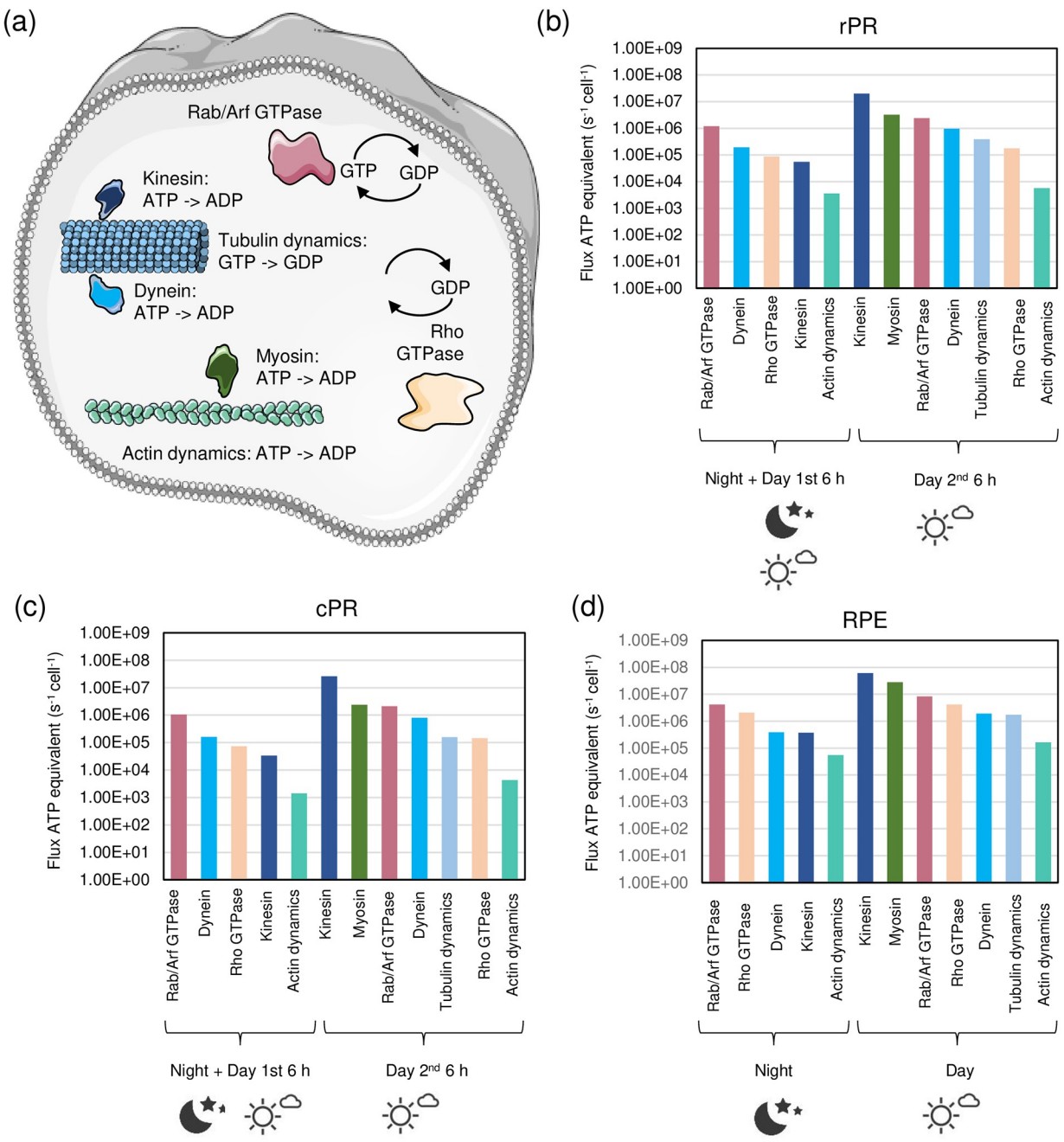

**Fig 2. Energy requirements for processes associated to the cytoskeleton.** (a) Visualization of key energy-requiring proteins involved in intracellular cytoskeleton processes. Shown are actin polymers and myosin motors in green, tubulin polymers and the tubulin-associated motor proteins kinesin and dynein in blue, Rab/Arf GTPases in red, and Rho GTPases in yellow. (b-d) ATP fluxes per cell related to cytoskeleton processes for rPR, cPR, and RPE at night and day. The Figure was partly generated using Servier Medical Art, provided by Servier, licensed under a Creative Commons Attribution 3.0 unported license.

associated with actin fiber polymerization and depolymerization, we calculated the total number of actin fibers based on protein abundances together with rates of association and dissociation (2.42 s$^{-1}$; Bionumbers ID: 107898). We assumed that basal actin remodeling mainly happens at night and the 1$^{st}$ 6-hour day-period in rPR and cPR (leading to ATP fluxes of $1.95 \times 10^3$ ATP/s/cell and $1.43 \times 10^3$ ATP/s/cell, respectively) and faster remodeling during the 2$^{nd}$ 6-hour day-period (leading to ATP fluxes of $5.79 \times 10^3$ ATP/s/cell and $4.26 \times 10^3$ ATP/s/cell, respectively) as cytoskeletal remodeling will be important during the anabolic phase (e.g. for organelle biosynthesis [35]).

Myosins are ATP-consuming motor proteins that move on actin filaments with their cargo (e.g. rhodopsin in rPR cells). The energy budgets were calculated using enzyme abundances and $k_{cat}$ (8 s$^{-1}$; [36]) values with additional constraints from availability of space on the total actin fiber network. Transport by myosins was assumed to mainly take place during the 2$^{nd}$ 6-hour day-period (anabolic phase) in rPR and cPR cells. For RPE cells, as myosins have functions in phagocytosis [37], we considered myosin-mediated transport in both 6-hour periods during the day. The flux of ATP through myosins is predicted to be in the range of $\sim 2 \times 10^6$ to $3 \times 10^7$ ATP/s/cell.

Tubulin fiber dynamics were calculated from the number of tubulin fibers and rates of polymerization and depolymerization. We assumed that tubulin dynamics is high for rPR and cPR during the 2$^{nd}$ 6-hour day-period (anabolism). As microtubules are important for vesicular transport (anabolism) and during phagocytosis of rod OS by RPE [38], we assumed that tubulin remodeling takes place during the 1$^{st}$ and 2$^{nd}$ 6-hour day-period in RPE cells. The flux of ATP (= equivalents of GTP) required for tubulin remodeling is predicted to be in the range of $\sim 3 \times 10^5$ to $2 \times 10^6$ ATP/s/cell.

Kinesins are ATP-driven motor proteins that move along microtubules. As kinesins are highly abundant and the maximal catalytic activity can be quite high (80 to 100 s$^{-1}$) [28, 39], (with a range between 0.5 and 80 s$^{-1}$; [39]) care in making assumptions is required to not over-estimate the budget. As with myosins, we estimated how many kinesins can fit on the total tubulin fiber network at different densities. At high densities, all kinesin molecules have space on the tubulin fiber network (S1 Fig in S1 File). At night (for all cells) and during the 1$^{st}$ 6-hour day-period (for rod and cone PR) we assumed a low density of kinesins and low catalytic activity (0.5 s$^{-1}$), which results in fluxes in the range of $\sim 6 \times 10^4$ to $4 \times 10^5$ ATP/s/cell. During the 2$^{nd}$ 6-hour day-period (and additionally for RPE in the 1$^{st}$ 6-hour day-period), we assumed intermediate numbers of kinesins on microtubules as, even during phases of high traffic, there should still some kinesin without cargo or with cargo but not on the microtubule. Together with medium catalytic activities (average of 0.5 and 80 s$^{-1}$ = 40.25 s$^{-1}$), this results in a budget in the range of $\sim 2 \times 10^7$ to $6 \times 10^7$ ATP/s/cell.

Dynein motors also walk on microtubules, but they are expressed $\sim$10-fold lower. The activities range between 0.75 and 15.18 (average = 8) s$^{-1}$ depending on the concentrations of microtubules [40]. With similar considerations as for kinesins (see also S1 File), the energy budget at night (for all cells) and during the 1$^{st}$ 6-hour day-period (for RPE) is in the range of $\sim 2 \times 10^5$ to $4 \times 10^5$ ATP/s/cell. For the 2$^{nd}$ 6-hour day-period (and additionally the 1$^{st}$ 6-hour day-period in RPE) the ATP fluxes through dyneins are between $\sim 8 \times 10^5$ and $2 \times 10^6$ ATP/s/cell.

Arf and Rab GTPases require energy by cycling between GDP- and GTP-bound states (with an estimated average rate of 0.3125 s$^{-1}$;[41]) and are involved in the cytoskeleton/vesicular transport [41, 42]. Hence, cycling activities are predicted in-line with actin- and tubulin-related processes. The predicted ATP fluxes during the night (for all cells) and the 1$^{st}$ 6-hour day-period (for rod and cone PR) are in the range of $\sim 1 \times 10^6$ to $4 \times 10^6$ ATP/s/cell. For the 1$^{st}$ 6-hour day-period in RPE and the 2$^{nd}$ 6-hour day-period (in all cells) the activities are higher with ATP fluxes between $\sim 2 \times 10^6$ and $8 \times 10^6$ ATP/s/cell.

Rho GTPases, also involved in cytoskeleton remodeling [43], were treated similarly to Arf and Rab GTPases (with an estimated average cycling rate of 0.3125 s$^{-1}$; [43]). The ATP fluxes during the night (for all cells) and during the 1$^{st}$ 6-hour day-period (for rod and cone PR) are between $\sim 7 \times 10^4$ and $2 \times 10^6$ ATP/s/cell. For the 1$^{st}$ 6-hour day-period in RPE and for the 2$^{nd}$ 6-hour day-period (in all cells) the activities are higher, with ATP fluxes of $\sim 1 \times 10^5$ to $4 \times 10^6$ ATP/s/cell.

Summarizing, the ATP fluxes of cytoskeleton-related processes span several orders of magnitude between $\sim 10^3$ to $\sim 10^7$ ATP/s/cell (Fig 2). During the night, cycling of Rab, Arf, and Rho GTPases contribute most. During the day, kinesins and myosins have the highest ATP fluxes.

**Protein turnover.**   Protein turnover was modelled based on demand for mRNA, ribosomes and the turnover of the proteins themselves (Fig 3).

mRNA turnover was estimated to be 9 h for all cells [44]. Ribosomes turnover has a half-life of 5 days in fasted rat liver [45]. Other studies suggest a half-life of 6 days for brain and 4 days for liver [46]. Here, we assumed a ribosome half-life of 6 days for rPR and cPR cells and of 4.5 days for RPE. Regarding protein half-life, for RPE cells we assumed 3 days [47]. As 1/10 of OS are phagocytosed every day [48], we assumed a turnover of 10 days for the proteins in rPR OS. For neurons protein half-life in culture is 5.4 days and 9 days *in vivo* [49]. As total brain includes myelin which contains proteins of very long half-life, we used a protein half-life of 5.4 days for the rest of the rPR and for cPR.

We assumed that anabolic processes take place mainly during the 2$^{nd}$ 6-hour day-period. To see if this is feasible as enzymes need to fulfill the same demand in 4 x the speed, we tested what key enzymes in transcription (RNA polymerase) and translation (ribosome) can maximally deliver in terms of biosynthesis. We find that based on enzyme abundance and k$_{cat}$ values the synthesis capacity exceeds at least by $\sim$24-fold what is needed by demand. Hence, assigning both transcription and translation to only a 6-hour time period is feasible.

We find in all three cell types that translation dominates the energy budget (Fig 3). This is followed by primary mRNA transcript production. These fluxes are very similar to the third highest contributor, which is lysosomal protein degradation (of half of the proteins and suggested to happen mainly during the catabolic phase of 1$^{st}$ 6-hour day-period). There are several energy-requiring steps in ribosome biogenesis [50], but production of ribosomal RNA by far dominates this term. Summing up all ATP fluxes in ribosome biogenesis results in a budget of $1.92 \times 10^5$ ATP/s/cell for rod and cone PR cells, and $1.50 \times 10^6$ ATP/s/cell (for RPE).

Proteasomal degradation also contributes, with ATP fluxes similar to what the proteasome can deliver based on cell-specific abundances and k$_{cat}$ (see S1 File). Therefore, we assumed that the proteasome is operating night and day in order to have the capacity to deal with the demand of proteins that must be degraded. The ATP fluxes for proteasomal degradation are between $\sim 8 \times 10^4$ and $7 \times 10^5$ ATP/s/cell.

Other energy consumers in protein turnover are posttranslational modifications (we considered phosphorylation, acetylation, glycosylation and ubiquitination), which we assumed to continually happen over the 24-hour cycle, folding via chaperones, and mRNA splicing/export.

**Lipid turnover.**   Cholesterol production was only considered in OS of the rPR as those are phagocytosed by the RPE cells. In the remaining part of the rPR, in cPR, and in RPE we assumed that cholesterol is largely recycled and is only a negligible energy term. Based on a demand of cholesterol production resulting from the daily shedding of OS and assuming that cholesterol is mainly produced during the 2$^{nd}$ 6-hour day-period, we calculated an ATP flux of $1.70 \times 10^6$ ATP/s/cell in 6 hours (Fig 4(a) and 4(b)).

The phospholipid turnover cycle was considered as breakdown into glycerol and free fatty acids and the energy-requiring reformation of phospholipid assumed to take place in the

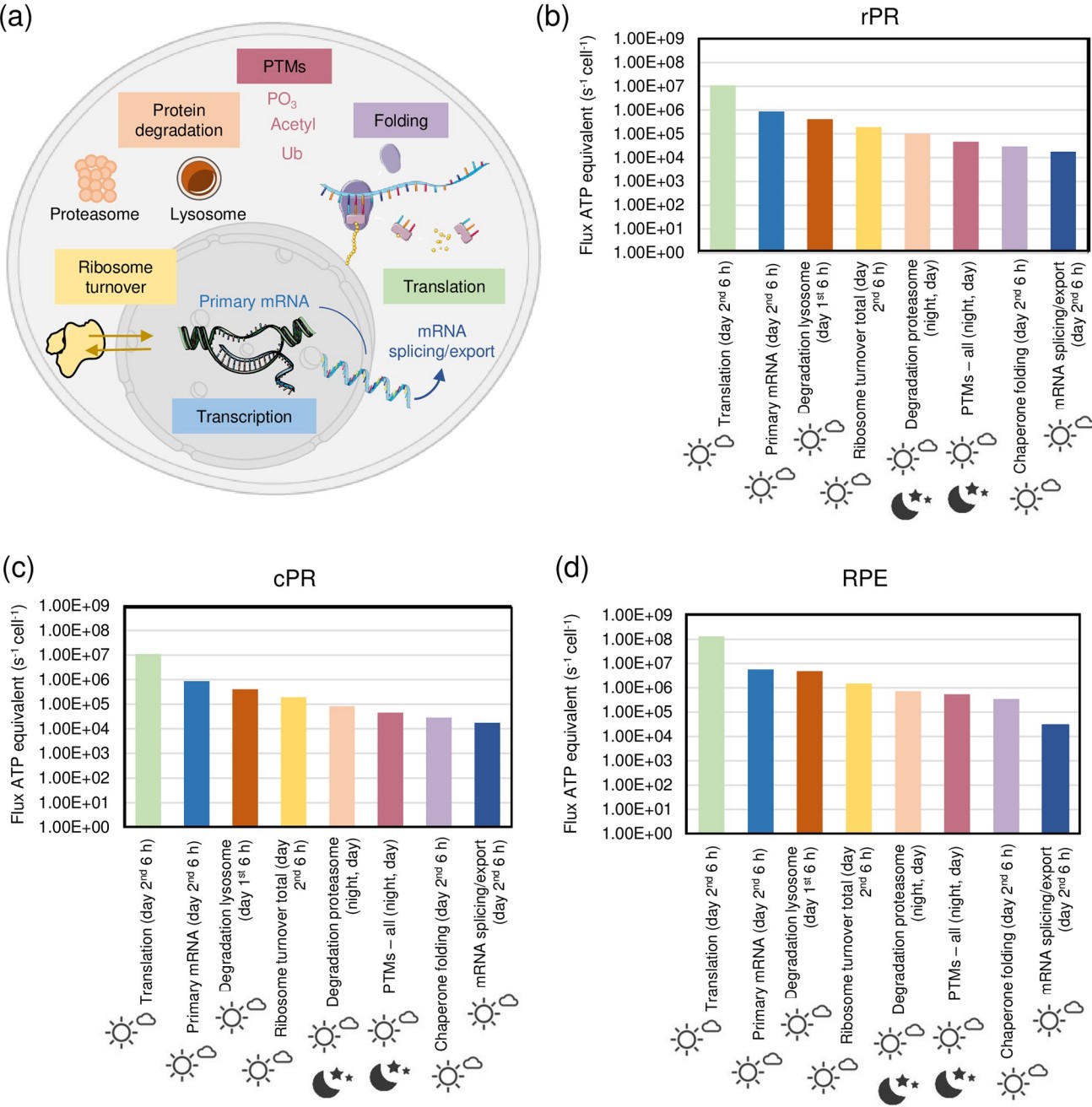

**Fig 3. Energy requirements for processes associated to protein turnover.** (a) Visualization of processes involved in protein turnover, which are transcription, translation, folding, posttranslational modifications, protein degradation, and ribosome turnover. (b-d) ATP fluxes per cell related to different protein turnover processes for rPR, cPR, and RPE at night and day. The Figure was partly generated using Servier Medical Art, provided by Servier, licensed under a Creative Commons Attribution 3.0 unported license.

anabolic $2^{nd}$ 6-hour day-period. Based on demand (a half-life of 4 days [51]), the associated ATP fluxes range between $\sim 6 \times 10^5$ and $4 \times 10^6$ ATP/s/cell (Fig 4(a)–4(d)).

With respect to mitochondrial turnover, we assumed that cholesterol is fully recycled and that phospholipid turnover mainly involves acyl chain remodeling and the energy for this is

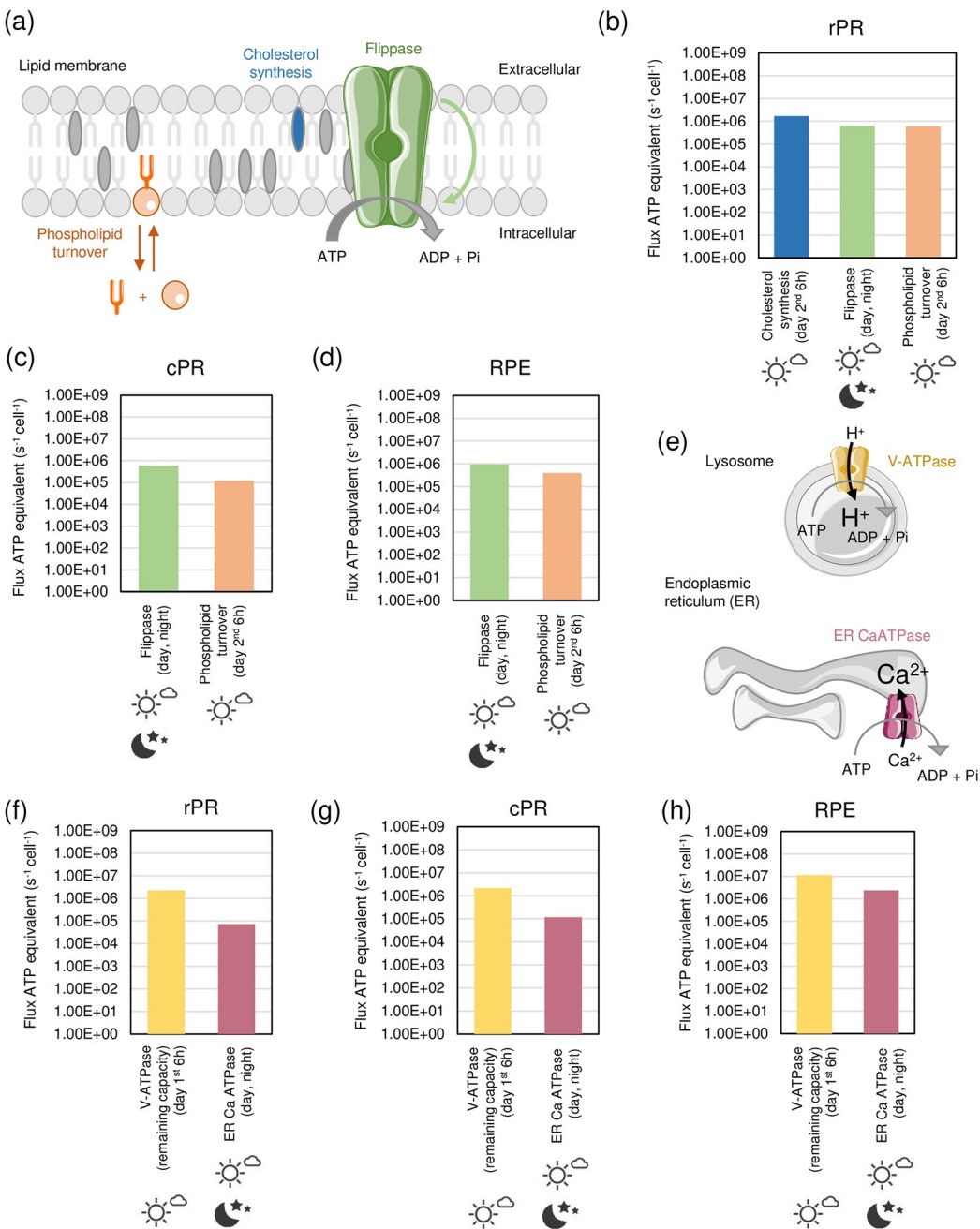

**Fig 4. Energy demand for lipid turnover and intracellular ion transport.** (a) Visualization of processes involved in phospholipid turnover, cholesterol synthesis and flipping of lipids in the PM. (b-d) ATP fluxes per cell related to lipid turnover in rPR, cPR and RPE cells in night and dark. (e) Schematic representation of processes linked to intracellular ion transport in lysosomes and ER. (f-h) ATP fluxes per cell for rPR, cPR and RPE cells in night and day, The Figure was partly generated using Servier Medical Art, provided by Servier, licensed under a Creative Commons Attribution 3.0 unported license.

already accounted for through total phospholipid turnover. Likewise, for the ER as the largest organelle in the cell, the half-life of ER lipid and proteins is estimated to be 3 to 5 days [51].

The P4-ATPases ("flippases") maintain membrane asymmetry by flipping phospholipids against a concentration gradient. Most flippases flip from cytoplasmic to exoplasmic leaflets

(such as the ABC transporters) but the P4-ATPases operate in the opposite direction. Photoreceptor OS express ATP8A2 and this will flip phosphatidyl serine (PS) from exo- to cytoplasmic leaflet and its activity is essential to maintain PR length (appearance of PS in the outer leaflet is a signal for phagocytosis and in the OS helps trigger tip shedding) and its absence, or even mis-localisation away from the OS, causes photoreceptor degeneration [52]. The energy required for maintaining cell membrane asymmetry was calculated based on the abundances of flippases and multiplied by their $k_{cat}$ of 0.5 s$^{-1}$ [53]. This is expected to happen continuously over 24 hours and ATP fluxes range between ~$1 \times 10^5$ and $4 \times 10^5$ ATP/s/cell (Fig 4(a)–4(d)).

**Intracellular ion transport.** The lysosomal vacuolar H+ ATPase (V-ATPases) is needed for lysosomal protein degradation and other degradative lysosomal functions [54]. The maximal capacity of this enzyme can be estimated from enzyme abundances in rPR, cPR and RPE together with $k_{cat}$ (10 s$^{-1}$) [55]. As we have calculated the demand of V-ATPases for protein degradation, the remaining capacity is available for other lysosomal functions and can be calculated from maximum capacity minus requirement of protein degradation by demand. This results in ATP fluxes between ~$2 \times 10^6$ and $1 \times 10^7$ ATP/s/cell (Fig 4(e)–4(h)). These values are likely upper limits as calculated from enzyme abundance and $k_{cat}$.

The leakage current across mitochondrial inner membrane seems quite large at 5 fA/mitochondria [56]. Assuming a thousand mitochondria for a typical cell [57], this equates to 5 pA or 3 x $10^7$ ions/s. Hence, this is the number of protons that are not available for ATP synthesis and could be converted to an ATP term as 2 H$^+$/ATP needed. To avoid double counting as there is a range of ATP production per glucose by oxidative phosphorylation (OxPhos) from 32 to 36, we considered throughout that only 32 ATPs per glucose are made via glycolysis and OxPhos. We like to point out the potential importance of mitochondrial uncoupling that we are not taking into consideration here [58]. It is suggested that photoreceptor mitochondria may be more uncoupled than in most other types of cells.

The endoplasmic reticulum (ER) Ca$^{2+}$-ATPase catalyzes the ATP-driven translocation of calcium from the cytosol to the ER lumen. The energy budget was calculated based on enzyme abundance and $k_{cat}$ and it was assumed to be a basic homeostatic process that continuously operates over 24 hours. The resulting ATP fluxes are in the range of ~$7 \times 10^4$ (rPR and cPR) to $2 \times 10^6$ (RPE) ATP/s/cell (Fig 4(e)–4(h)). These values are likely upper limits as calculated from enzyme abundance and $k_{cat}$.

**DNA repair.** Starting from the different types of DNA damage the average number of events per day per cell was estimated [59–63], which resulted in a total DNA damage rate of $3.75 \times 10^4$ events/day/cell. Next, we collated information about the amount of ATP required for the repair of the different types of DNA damage. The total energy requirement for DNA repair is only minor with 45.06 ATP/s/cell and 0.66 Glucose/s/cell for the three cell types. This process is predicted to be mainly active in the 2$^{nd}$ 6-hour day-period as data in mouse (which are nocturnal animals and in this respect like the retina have their highest energy demand in the night) show that nucleotide excision repair activity is highest in the afternoon/evening hours [64].

**OR-specific cell processes.** The following retina-specific processes were considered for the energy budget: Dark current in rPR (already covered in the section of PM ion transport), visual transduction in rPR, retinal recycling in rPR and RPE, synaptic transmission in rPR, cPR (and Mueller cell), and phagocytosis of OS by RPE cells (which is covered in the section of cytoskeleton dynamics).

In rPR cells the largest energy demand ($1.00 \times 10^6$ ATP cell$^{-1}$ s$^{-1}$) during day is for PDE6D (retinal rod rhodopsin-sensitive cGMP 3',5'-cyclic phosphodiesterase subunit delta) activation and the associated guanylate cyclase activity (for the recycling of GTP) (S2A Fig in S1 File). In the night, similar high ATP fluxes are predicted for the energy required for glutamate recycling

because of synaptic activity. Minor energy budget items related to visual transduction in rPR cells are transducin activation, retinal recycling and rhodopsin phosphorylation. In cPR cells, $2.28 \times 10^6$ ATP/s/cell are required for glutamate recycling for synaptic transmission at night, $1.0 \times 10^6$ ATP/s/cell for guanylate cyclase activity under indoor light condition (S2B Fig in S1 File). In RPE cells, $2.38 \times 10^5$ ATP/s/cell are needed for retinal recycling during indoor light conditions (S2C Fig in S1 File).

## Energy budget per cell type

**Rod photorecepor cell.** A comparative analysis of the highest ATP fluxes per second per rPR cell shows the dominating processes at night, which are the dark current and the energy required for maintenance of calcium ion gradients in the dark (S3 Fig in S1 File). This is directly followed by kinesin motor activities (2nd 6-hour day-period), the energy for translation (2nd 6-hour day-period), basal sodium-potassium ATPase activity (daytime), myosin motors (2nd 6-hour day-period), Rab and Arf GTPases (2nd 6-hour day-period), lysosomal activities other than protein degradation (1st 6-hour day-period), basal calcium ATPases (daytime), cholesterol production (2nd 6-hour day-period), Rab and Arf GTPases (during night and 1st 6-hour day-period), and PDEdelta activation and guanylate cyclase activity (during daytime).

Another way to represent the energy budget is to sum sub-processes involved in the eight main classes of cellular process (PM ion transport, cytoskeleton, protein turnover, lipid turnover, intracellular ion transport, synaptic transmission, visual transduction, and DNA repair; see legend of Fig 5) and integrate daytime processes happening during one or both of the 6-hour time-periods into a 12-hour daytime frame. This shows that in the rPR in the night energies are strongly dominated by the dark current (Fig 5(a) left). Other energy requiring processes in the night are cytoskeleton, lipid turnover, and synaptic transmission (Fig 5(a) left insert). During the day, highest contributions are from the cytoskeleton, followed by PM ion transport, protein turnover, lipid turnover, intracellular ion transport and visual transduction (Fig 5(a) right).

**Cone photorecepor cell.** Comparing the highest ATP fluxes per second per cPR cell reveals that the dominating processes at night are the ATPases involved in $Ca^{2+}$ (night and day) and $Na^+/K^+$ transport (during night) (S4 Fig in S1 File). Slightly lower are the energies required for kinesin motors (2nd 6-hour day-period), followed by $Na^+/K^+$-ATPase daytime activity, translation (2nd 6-hour day-period), myosins (2nd 6-hour day-period), glutamate recycling (nighttime), lysosomal activities other than protein degradation (1st 6-hour day-period), Rab and Arf GTPases (2nd 6-hour day-period), Rab and Arf GTPases (during night and 1st 6-hour day-period), and daytime guanylate cyclase activity.

Comparing the energy sums across the main eight energy-requiring processes in the cell shows that in the night the dominant energy-consuming process is PM ion transport (Fig 5(b) left). With respect to the smaller contributing processes in the dark, synaptic transmission dominates, followed by cytoskeleton processes, while lipid and protein turnover represent small budget items during the night. During the day, in contrast to rPR cells, PM ion transport energy fluxes are still dominating in cPR cells (Fig 5(b) right). This is followed by cytoskeleton processes and protein turnover, and other minor contributing terms.

**Retinal pigment epithelium cell.** In RPE cells, the two highest energy requiring processes are protein translation and the activity of $Na^+/K^+$-ATPases (S5 Fig in S1 File). However, translation is predicted to take place only during the 2nd 6-hour day-period, while the maintenance of $Na^+/K^+$ gradients is a 24-hour process. This is followed by ATP fluxes needed for kinesins (daytime), myosins (daytime), lysosomal activities other than protein degradation (1st 6-hour day-period), $Ca^{2+}$-ATPases (daytime), Rab and Arf GTPases (daytime), primary mRNA

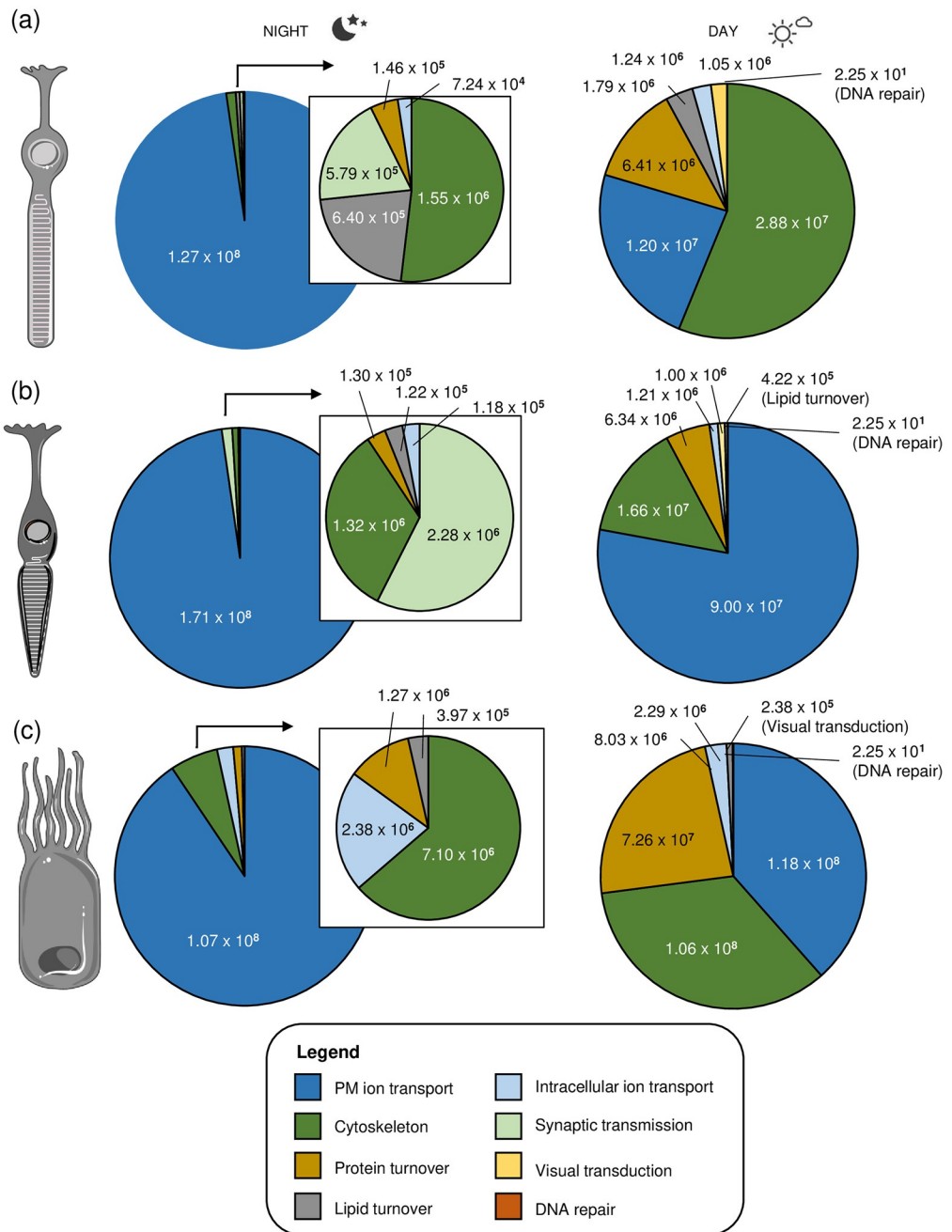

**Fig 5. Summary of the energy budgets classified into eight processes for rPR, cPR and RPE cells.** (a-c) Pie charts show fractions of energies summed for sub-processes in the eight main energy classes, which are PM ion transport, cytoskeleton, protein turnover, lipid turnover, intracellular ion transport, synaptic transmission, visual transduction, and DNA repair. The inserts next to the pie charts at night represent all processed except the dominating energy requiring PM ion transport.

production (2nd 6-hour day-period), lysosomal protein degradation (1st 6-hour day-period), Rab and Arf GTPases (nighttime), Rho GTPases (daytime), calcium ATPases in the ER (night and day), Rho GTPases (night), dyneins (daytime), tubulin dynamics (daytime), and production of rRNA (2nd 6-hour day-period).

With respect to the eight main cellular energy-requiring processes, during the night PM ion transport is the dominant process (Fig 5(c) left). This is followed by cytoskeleton processes, intracellular ion transport and protein turnover. During the day there is a roughly equal distribution of energies needed for PM ion transport, cytoskeleton processes and protein turnover (Fig 5(c) right).

## Energy budget per mm$^2$ tissue

To analyse the energy consumption per unit area of OR, two regions were considered: one (perifoveal) rod-dominated 1 mm$^2$ patch and one cone dominated (foveal) 1 mm$^2$ patch (Fig 6(a)–6(c)). This enabled comparison with ATP supply estimates in the companion paper (Fig 6(d); Prins et al 2024 [65]) but, as the latter rates were calculated on the basis of oxygen consumption by the OR, we subtracted, from the per cell energy budget described here, the ATP required to maintain the synapse. The largest component of this, especially for cones, is the PM Ca$^{2+}$-ATPase. From several immuno-histochemical studies, most PM Ca$^{2+}$-ATPase is localised predominantly to the synapse, although some appears to be present at the IS. For the present purposes we distributed 80% to the synapse and 20% to the IS.

As expected, the highest energy demands are predicted during the night in the rod-dominated tissue patch at 1.69 x 10$^{13}$ ATP/s/mm$^2$ tissue. In the day this drops by about half to 6.53 x 10$^{12}$ ATP/s/mm$^2$ tissue. In the cone-rich fovea, the night-time figure is 6.41 x 10$^{12}$ ATP/s/mm$^2$ tissue with a modest rise to 6.75 x 10$^{12}$ ATP/s/mm$^2$ tissue in the day. Of note, energy demands in RPE cells are increasing during the day (1.23 x 10$^{12}$ ATP/s/mm$^2$ tissue), which corresponds to an increase from 3.6% RPE (night) to 18.8% (day) energy requirements in a rod-dominated tissue and an increase, respectively, of 7.4% and 18% in a cone dominated tissue.

## Discussion

Like the brain, the OR has an exceptionally high energy expenditure. Here, we provide a fine-grained energy budget for three key cell types in the human OR. By taking a systems biology approach, we have quantified ATP (or ATP equivalent) fluxes through ~40 cellular subprocesses on that require energy. Both, cell general (housekeeping) and OR-specific processes were considered. Our predicted ATP expenses compare well to a previous study by Okawa et al 2008 in rod photoreceptors [1]. In the dark, we predict a flux of 1.09 x 10$^8$ ATP s$^{-1}$ rod$^{-1}$ compared to 1 x 10$^8$ ATP s$^{-1}$ rod$^{-1}$ by Okawa et al, 2008). In indoor light conditions, we predict a flux of 3.53 x 10$^7$ ATP s$^{-1}$ rod$^{-1}$, which is slightly higher (as expected) as predicted in saturating light illumination intensity (<2.50 x 107 ATP s-1 rod; [1]).

We predict a high energy budget for cytoskeleton-related processes, such as kinesin molecular motors. For example, transport of rhodopsin and other proteins along the cilium to the OS involves kinesins; and loss of those proteins can cause PR cell death [66]. Likewise, many processes in RPE cells (e.g. intracellular trafficking and phagocytosis) are linked to highly organized and polarized cytoskeleton components, which are often perturbed under stress conditions or in diseases such as age-related macular degeneration [67].

We used two approaches to assess the energy budget: based on demand and based on enzyme abundance and k$_{cat}$. We were able to compare the two alternative approaches for PM ion transport. We found that the Na$^+$/K$^+$ ion transport estimated by demand agreed well with those calculated from Na$^+$/K$^+$ transporter activity and k$_{cat}$. In this instance, the enzymes would be predicted to work at their maximal capacity, especially in the dark. In contrast, calcium ion transport demand generally seems to provide a lower estimate in comparison to rates based on transporter abundance and k$_{cat}$. Likewise, for some subprocesses within protein turnover we

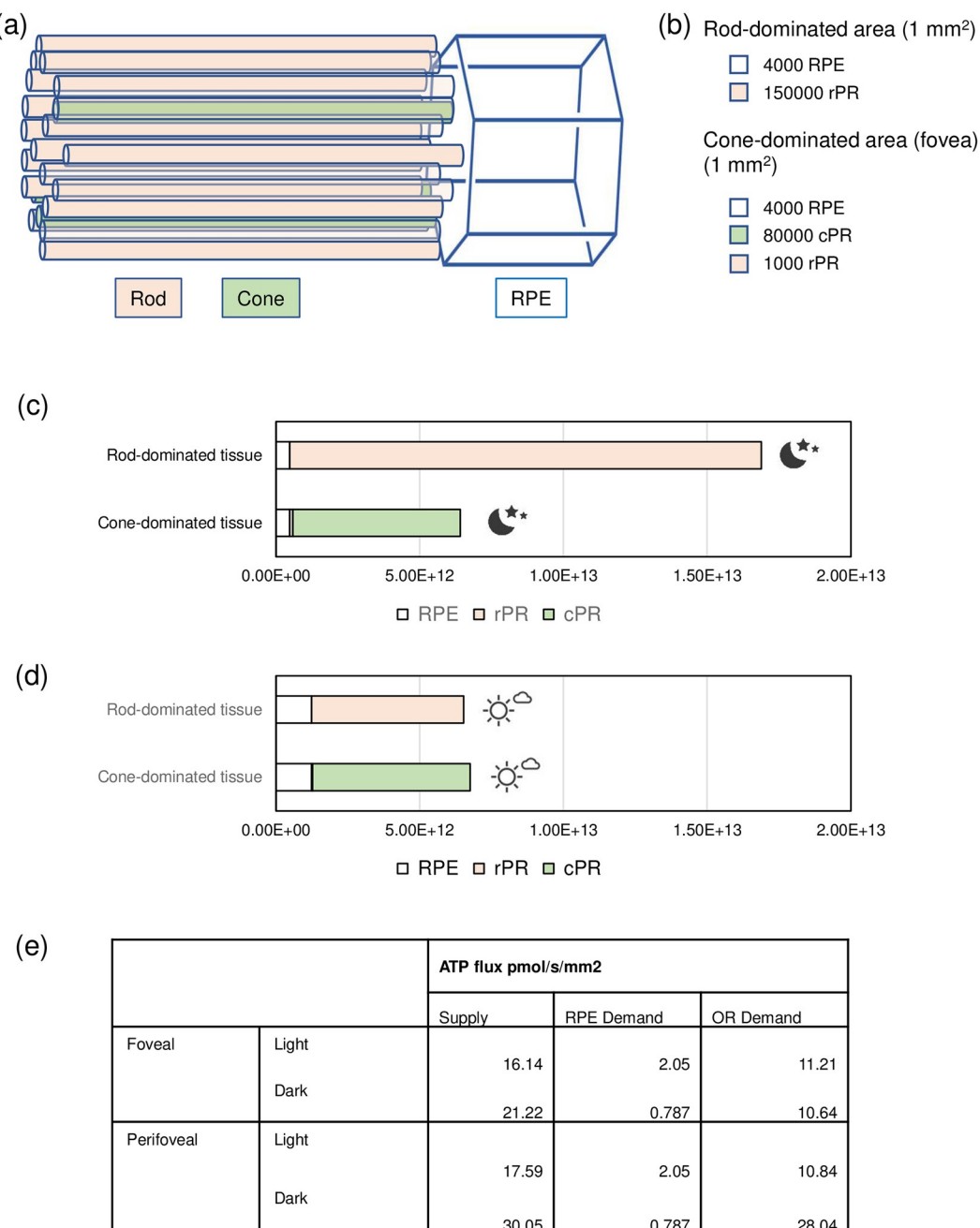

**Fig 6. Predicted energy consumptions in rod- and cone-dominated outer retina areas.** (a) schematic representation of approximate relation in number of photoreceptors to rpe cell. (b) Number of cells in rod- or cone-dominated 1 mm² tissue of outer retina considered for the calculations of energy consumption rates in (c) and (d). (c) Predicted ATP consumption in rod or cone dominated tissue during the night (in ATP molecules/s/mm² tissue). (d) Predicted ATP consumption in rod or cone dominated tissue during the night (in ATP molecules/ mm² tissue/ sec). (e) Comparison of ATP supply and demand estimates. Supply data refer to combined RPE and cone model for fovea and combined RPE and rod model for perifovea (see accompanying paper by Prins et al [65]).

were able to compare calculations based on demand with those based on enzyme abundancies and $k_{cat}$. We found that both the capacity of RNA polymerase and of the ribosome is higher (between 24 to 150-fold) than what appears needed by demand. This makes biological sense as it would still enable transcription and translation to occur in bursts for a short period of time.

Another important limitation is the use of mRNA data from single cell RNAseq as a proxy for protein levels. Correlation coefficients between mRNA and protein abundance are < 0.7–0.8 [19]. As mRNA and protein abundances have a high dynamic range, despite relatively high correlations, there might be individual proteins that have significantly higher or lower protein levels as estimated from mRNA. However, in the absence of quantitative single-cell proteomic data, there is no real solution to this problem.

The analyses and predictions carried out here required us to take many assumptions (which are detailed in S1 Text in S1 File). Particularly, the temporal assignment of catabolic and anabolic processes into the 1$^{st}$ or 2$^{nd}$ 6-hour day-period is an assumption. However, the assumption that some house-keeping tasks are preferentially delivered during the day seems plausible considering that energy flows in the dark are maximised to meet ion transport demands. Indeed our hypothesis is supported by the observation that the difference between metabolic fluxes at night and day is significantly less than that anticipated from the difference in dark current [68]. Furthermore, rod OS are shed and phagocytosed early in the day. These will have to be broken down prior to recycling and re-synthesis into macromolecules for the photoreceptors later in the day so there are probably metabolic shifts during the day from catabolism to anabolism.

Estimates of ATP supply (see companion paper; Prins et al 2024 [65]) exceed those for ATP demand by, on average, a factor of 1.6. Some of the possible explanations are discussed in the companion paper. With respect to the energy demand prediction, there are some aspects to be explored in more detail. Particularly, cytoskeleton-associated processes have potentially been under-estimated as generally lower limits or averages were used in the calculations (e.g. for the occupancy or movements of molecular motors along tubulin and actin fibres). Hence, those processes may happen at higher rate if needed. Both, RPE cells and photoreceptors are expected to heavily rely on cytoskeleton support to normal physiological functions, such as phagocytosis, intracellular trafficking, and maintaining cell shape and geometry [67, 69]. Also, the number and volume of cells and the relative size of different cellular compartments differs depending on the region of the retina. This could cause errors in the energy demand predictions for a specific 1 mm$^2$ area considered here. In this respect, this study represents a starting point and when more quantitative anatomical, biochemical and physiological data become available in the future, the numbers provided here can be updated. This seems particularly important where currently some predictions are based on data obtained from non-human or non-retinal (e.g. liver) tissues.

In recent years there has been much interest in age-related changes in the bioenergetics of this system. As considered in the companion study (Prins et al 2024 [65]) in normal aging there appears to be a reasonable balance between degradation of the choriocapillaris vascular network and loss of rPR. Aging of the OR is, however, also associated with other changes including thickening of Bruch's membrane [70] and the formation of diffuse and focal sub-RPE deposits. Energy failure in the RPE is hypothesised to be a possibly important element in the pathogenesis of AMD and it is interesting that our estimates of RPE energy use in the light is such a high percentage of the OR total. We know little of how cells balance energy demand of different processes but even with the reduction in dark current demand in daytime it is plausible that inability to meet substantial house-keeping energy requirements contribute to age-related, blinding diseases such as age-related macular degeneration. In line with this, we

propose that in the future the possibility of tuning the energy budget exists, e.g. by lowering some less critical cytoskeleton processes to save energy.

## Supporting information

**S1 File. Contains S1 Text and S1 to S5 Figs.**
(PDF)

**S1 Table. Molecular census per cell.**
(XLSX)

**S2 Table. Gene expression levels in RPE, rPR, and cPR cells and estimates for number of proteins in cells and protein concentration.**
(XLSX)

**S3 Table. Currency conversions.**
(DOCX)

**S4 Table. Detailed energy budget in rPR, cPR, and RPE cells.**
(XLSX)

## Author Contributions

**Conceptualization:** Christina Kiel, Alexander J. E. Foss, Philip J. Luthert.

**Data curation:** Christina Kiel, Alexander J. E. Foss.

**Investigation:** Christina Kiel, Stella Prins, Philip J. Luthert.

**Methodology:** Christina Kiel, Alexander J. E. Foss.

**Writing – original draft:** Christina Kiel, Alexander J. E. Foss.

**Writing – review & editing:** Stella Prins, Alexander J. E. Foss, Philip J. Luthert.

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
