## [Decision Letter · Decision Letter 0]

17 Jul 2024

PONE-D-24-23593Energetics of the outer retina II: Calculation of a spatio-temporal energy budget in retinal pigment epithelium and photoreceptor cells based on quantification of cellular processesPLOS ONE

Dear Dr. Kiel,

Thank you for submitting your manuscript to PLOS ONE. After careful consideration, we feel that it has merit but does not fully meet PLOS ONE’s publication criteria as it currently stands. Therefore, we invite you to submit a revised version of the manuscript that addresses the points raised during the review process.

All reviewers state that your manuscript merit publication. However, all of them see also the need to improve the calculations of the model:. Issues such as saturation of transporters, collateral transportation pathways for Na+ and the role of ATP require improvements. 

We look forward to receiving your revised manuscript.

Kind regards,

Olaf Strauß

Academic Editor

PLOS ONE

Journal Requirements:

 "This work was supported by a Moorfields Eye Charity (Grant GR001345)."

Reviewers' comments:

Reviewer's Responses to Questions

**Comments to the Author**

1. Is the manuscript technically sound, and do the data support the conclusions?

Reviewer #1: Yes

Reviewer #2: Partly

Reviewer #3: Yes

2. Has the statistical analysis been performed appropriately and rigorously? 

Reviewer #1: Yes

Reviewer #2: N/A

Reviewer #3: Yes

3. Have the authors made all data underlying the findings in their manuscript fully available?

Reviewer #1: Yes

Reviewer #2: Yes

Reviewer #3: Yes

4. Is the manuscript presented in an intelligible fashion and written in standard English?

Reviewer #1: Yes

Reviewer #2: Yes

Reviewer #3: Yes

5. Review Comments to the Author

Reviewer #1: Review for PONE-D-24-23593 - Energetics of the outer retina II: Calculation of a spatio-temporal energy budget in retinal pigment epithelium and photoreceptor cells based on quantification of cellular processes, Kiel et al., 2024

The manuscript by Kiel and colleagues tries to give detailed estimates of the energy consumption and the related cellular processes for the key cell types of the outer retina, namely retinal pigment epithelial (RPE) cells, and rod and cone photoreceptors. They combine gene expression data with biochemical knowledge to infer the ATP expenditure required for all major cellular processes and thereby obtain an unprecedented overview of outer retina energy consumption. Notably also because of its relevance for retinal disease pathogenesis, the topic of this article is highly attractive and appears well suited for PLoS One.

The manuscript is well written and the rather complex topic at hand is easy to follow. The figures shown generally are of high quality, with only minor shortcomings. However, some parts of the results may have been oversimplified and the authors appear to have missed the key role of Na+/K+/Ca2+ exchanger (NCKX) in regulating photoreceptor Ca2+ levels. These and other (mostly minor) issues must be appropriately addressed in a revised manuscript (see below).

Major points:

1. Results: The authors mention the key importance of quantitative data for their calculations and they employ gene expression data for their estimates of protein expression. This is problematic as gene expression and protein expression may not correlate well with each other, especially in low abundance transcripts. Here, even minor differences could lead to major distortions. For instance, the abundance of NKX inferred from gene expression could be an underestimation. While in the absence of quantitative single-cell proteomics there is no real solution to this problem, the problem as such and the possible misinterpretations (or limits to the current interpretation) should be mentioned clearly and appropriately discussed.

2. Results: In several instances the assumptions and calculations appear to have been oversimplified or are not properly presented. For example, the authors state that rods and cones “have an inward Ca2+ current in the dark” (line 196) without saying what the current is that they used for their subsequent ATP-flux calculations. This is similar for most parts of the results (e.g. for Na+ influx, actin fiber polymerization and depolymerization, etc.) where the key values and assumptions that were used to make calculations should always be given in the main text.

3. Results: In the first part the authors suggest that ATP consumption from NKX activity was on par with or even lower than that from PMCA activity. This is in contradiction to earlier works that found NKX to consume at least 50% of the total photoreceptor ATP production (e.g. Ames et al., J Neurosci 12:840-853, 1992) and is somewhat counterintuitive given that the CNG-channel allows mostly for Na+ influx (only about 15-20% of the CNG current is carried by Ca2+).

Importantly, the authors seem to have overlooked that most of the Ca2+ influx mediated by the CNG-channel is compensated for by the Na+-gradient driven Na+/K+/Ca2+ exchanger (NCKX) in the outer segment (exchange of 1x Ca2+ and 1x K+ for 4x Na+). Hence, overall the Na+ influx and K+ outflux should be dramatically higher than the net Ca2+ influx, and, correspondingly, ATP-consumption for Ca2+ extrusion would be expected to be at least 1 order of magnitude lower than that for (re)establishing Na+/K+ gradients (through NKX activity).

The importance of NCKX for Ca2+ in the outer segment is also mirrored by the distribution of plasma membrane Ca2+-ATPase (PMCA) mostly to the synapse (something that the authors mention in the discussion). Without NCKX, PMCA would need to be far more strongly expressed in the inner segment to prevent Ca2+ from interfering with protein translation and gene expression.

The role of NCKX for photoreceptor Ca2+ levels must be acknowledged in the manuscript and the corresponding calculations should be revised.

4. All figures: Why are there no error bars in the bar graphs? The various measurements/estimates surely will have errors (or upper and lower boundaries) attached to them, these should also be included of the visualization of ATP fluxes.

5. Figure 1b: In my version of the manuscript, the labels of the bar graphs are cut off and only partly visible.

6. Figure 5: The pie charts should include numerical values for the different pie parts.

7. Figure 6: The authors should add a scaling to the x-axis, and perhaps vertical dashed lines, to more clearly indicate what the energy expenditure of the different tissues is.

Minor points

1. Abstract: The following sentence should probably be revised: “We propose the likely need of for diurnal…”.

2. Line 109: double plural: “…copies for each proteins was…”

3. The scaling of ATP flux in the various figures is variable, going from 1.00E+00 to 1.00E+07 to 1.00E+09. To allow an easier comparison across figures it would be helpful if the same logarithmic scale was used throughout all figures.

Reviewer #2: This is a wonderful effort and is of value to the scientific community. Too few investigators work to integrate information from many labs into a coherent analysis, so I very much appreciate efforts such as this one.

While I believe that a future form of this is worth publishing, have quibbles about some of the details in the manuscript’s current state. These are mostly geared towards using language that emphasizes the limitations of the manuscript to a greater extent. Some of the limitations are also due to specific assumptions that are difficult to believe would hold up realistically. These are noted with other comments below.

(1) By using turnover number (Kcat) and a transcriptome-based assessment of enzyme levels (line 92), two major assumptions are made:

a. The transcriptome and proteome correspond well to each other. This is not usually the case (PMIDs: 28438116, 20023718, 19850042). The concept that calculations of protein abundance or distribution could be incorrect is not discussed and should be, since this data is foundational to the rest of the study.

b. A Kcat is determined for saturating concentration of the enzyme’s substrate (here, often ATP). It seems unlikely that all enzymes simultaneously operate at maximum flux. This is a major assumption that would be questioned by many people if stated this way. Overall I do not think this study quantifies normal cellular energy budgets per se, rather it calculates maximum possible energy demands. The assumption that all ATPases would need to work at their Kcat is mentioned but not as a major limitation in the discussion. It should be. Additionally I recommend but will not require that the authors refer to their calculations as “maximums” calculated for a given cellular process, so that readers will not accidentally interpret the maximum possible ATP expenditures as physiological ATP expenditures. An example of this would be to transform the statement “we have quantified ATP (or ATP equivalent) fluxes” � ““we have calculated maximum possible ATP (or ATP equivalent) fluxes”. I think this language would make the meaning of this data more clear to readers.

(2) Line 67-69: Are these lengths and widths of inner and outer segments arbitrary or derived from a specific source in the literature? Regardless of where it originated from, it would be best to refer to the source of information used for cell compartment volume calculations.

(3) Line 141: So many of the downstream measurements depend on cell volume. That cone photoreceptors are assumed to be the same volume as rods with no more evidence than they are shorter and also wider seems ridiculous and calls into question the analytical rigor of a paper that is incredibly reliant on a solid analytical approach to data analysis. The authors should cite evidence for their determination of cone volume or provide an analysis of how estimated cell volume impacts energy utilization measurements.

(4) Line 143: There are differences in RPE surface area and presumably volume as a function of retinal eccentricity. This means that in rod-rich areas of the retina vs. cone-rich areas, there will be distinct RPE cell volumes, which could throw off rod-RPE-ratio estimates. This should be mentioned as a confounding factor in the discussion.

(5) Line 278: The citation used to rationalize a 3-day protein half-life in RPE appears to be geared towards the liver. RPE and liver are of course different tissues with different needs, as are RPE and photoreceptors. This is one example of where values from a different tissue are substituted for values of a similar process in photoreceptors or RPE. The practice seems necessary for this manuscript in the absence of better data, but a statement in the discussion should note that values for the rate of a process from distinct tissues and species may not be true when estimating the activity of that process for the human eye.

(6) Line 453: Do the authors mean “OR” or “IS” instead of “OS”?

(7) Line 505: The last word in the sentence should be “anabolism”

(8) Line 517: No year on this citation

(9) There is no propagation of experimental error from original sources of data. In an ideal world, estimates of ATP expenditure by each cellular process would have error bars representing uncertainty that was propagated from the original papers. I realize this last point is perhaps too much to ask so this modification is an optional request that would enhance the manuscript but will not be required from me for publication.

(10) To an extent the authors change the Kcat for different circadian periods. These changes often appear to be arbitrary and values are not always linked to a citation. All Kcat values should either be cited in text, or a citation should be linked to the supplemental spreadsheets. As the study stands some of the values seem to be cited but not all of them.

(11) Improvements in technology means that better retina, photoreceptor, or RPE-specific estimates of turnovers, PTMs, cytoskeletal movements, etc. might be published in the future. This study provides a framework for estimating energy expenditures that should ideally be updated as improved values for each process are determined experimentally. It is certainly not a requirement for the manuscript and is far outside my purview as a reviewer, but I highly recommend the authors consider generating a publicly available resource that uses the framework behind this study but can be updated. A web-based tool where people can deposit recommended values for a given process might be best. Such a strategy could drastically improve the usefulness and long-term accuracy of the current study.

Supplementary document:

• There are several instances in this section where assumptions are stated but the reasons for a given assumption are unstated. This leads to the reader (me) having the uneasy feeling that a substantial proportion of the calculations are determined using arbitrary starting values. In some cases the choice of constants could inform major outputs in this paper. Where possible, could the authors improve their explanations for a given assumption? This issue was noticeable in the following sections of the supplementary information: “We also assumed a higher kcat (average of 0.5 and 80 s -1 = 40.25 s-1 )”, “We initially assumed that 50 % of the proteins are degraded via the proteasome and 50 % via the lysosome”, “”

• The reference section has been added twice to this document

• For some information (for example the energetics costs from using Ca2+ ATPases), there is a range of possible ATP consumption values based on estimates and on published data. Yet in figures a single value is chosen to represent that energetic cost. Would it be feasible to represent ATP costs as a range of possible costs, where applicable? And not just for this section but wherever there is a known range of uncertain values in the calculation?

• The calculation of energetic costs associated with PTMs appear to be particularly unrealistic for a few reasons: (1) they assume every site on a protein that can be modified with a PTM is normally modified. It is normal in literature to see changes in the abundance of various PTM under different conditions, which indicates that not all possible PTM sites are normally occupied. This study would be improved by the incorporation of a term which quantifies the proportion of PTM sites that are normally occupied. (2) I think the ATP-equivalent values for production of molecules like acetyl-CoA are calculated incorrectly. The values used actually correspond with the energy that would be produced if acetyl-CoA were consumed. Use of as a PTM does not use up the energy it does not generate. The calculation should instead have the cost of Acetyl-CoA synthesis. Acetyl-CoA synthesis shares most if its reactions with glycolysis, but should additionally reduce a molecule of NAD+ to NADH. This of course assumes there is no limitation on CoA availability. When the production of a molecule used to modify a protein is energetically productive it should be subtracted from the energetic budget rather than added to it. (3) Acetylation, glycosylation, phosphorylation, and ubiquitination are far from the only posttranslational modifications. If the authors plan to only determine the energy budget associated with only these modifications, they should provide a reasonable explanation for why the other modifications are not taken into account. Perhaps other modifications are far less abundant on proteins? If so, please find and cite this information.

• Page 19: Estimates of protein turnover may be slightly off. This paper (PMID: 7452249) is an estimate of protein synthesis for the whole rabbit retina, but average protein synthesis rates in photoreceptors are expected to be faster than for other retinal cells. If I’ve calculated correctly from this paper, the estimate for the average retina protein half-life is almost exactly 4 days. Please consider using retina-specific estimates of turnover where possible.

• Page 21: I think “24 night and dark cycles” is meant to mean “24-hour light and dark cycle”

Reviewer #3: Strengths.

• This paper presents a comprehensive list of likely energy demands on photoreceptors and RPE cells. Then accompanying paper presents a corresponding list of the ways that fuels can be supplied to these cells and how the energy can be extracted from them and stored in ATP. The authors obviously have put a huge effort into bringing all this information together in these lists.

• The comprehensive nature of this study can be a starting point for quantitative modeling of energy supply and demand in the retina and RPE. Most of the data used are not really relevant for these ocular tissues, but more relevant data can be updated into this model as the authors and others recognize the usefulness of data from direct physiological and biochemical measurements that have been made and will become available on human and mouse ocular tissues and tissues from other species.

Weakness.

• The data used for these models in many cases are not appropriate or relevant. For example, it cannot be accurate to predict enzymatic activities from transcript levels. Translation of the transcripts is different for each protein, the stability of each enzyme is different and the enzymes are regulated by posttranslational modifications. For example light activated rhodopsin is inactivated by phosphorylation and there is so much rhodopsin in a rod that substantial amounts of ATP can be consumed in this process. That is an example of an energy demand that is unique to photoreceptors. The locations of enzymes in photoreceptors can change in light and darkness. These factors are relevant but not sufficiently considered in this model.

Overall, this is a useful starting point for this type of analysis and I recommend that the authors emphasize the purpose of this analysis in this way – that future analyses will incorporate actual data from biochemical and physiological studies of retina and RPE tissues into their model.

The authors should address a few specific comments and suggestions.

1. Lines 303-306: Rhodopsin phosophorylation is a major consumer of ATP. It only happens in light but there may be continuous phosphorylation and dephosphorylation when light is being absorbed.

2. In the discussion section please make a direct comparison of the current findings and conclusions with the estimates of ATP consumption that were reported by Okawa et al in 2008.

3. lines 347-353. It would be helpful to discuss the importance of mitochondrial coupling. Photoreceptor mitochondria may be more uncoupled than in most other types of cells. The authors cite reference 46, which is based on measurements of mitochondria from cultured cells and from mitoplast preparation. The properties of mitochondria can vary in the degree to which they are uncoupled and photoreceptor mitochondria may be particularly unique in this way. This should be discussed.

4. line 378 - please define what PDE delta is.

5. line 42- delete "on"

6. PLOS authors have the option to publish the peer review history of their article (what does this mean?). If published, this will include your full peer review and any attached files.

Reviewer #1: No

Reviewer #2: **Yes: **Daniel T Hass

Reviewer #3: **Yes: **James Hurley

---

## [Author Response · Author response to Decision Letter 0]

23 Aug 2024

-A rebuttal letter that responds to each point raised by the academic editor and reviewer(s). You should upload this letter as a separate file labeled 'Response to Reviewers'. OK

-A marked-up copy of your manuscript that highlights changes made to the original version. You should upload this as a separate file labeled 'Revised Manuscript with Track Changes'. OK

-An unmarked version of your revised paper without tracked changes. You should upload this as a separate file labeled 'Manuscript'. OK

---

## [Decision Letter · Decision Letter 1]

16 Sep 2024

Energetics of the outer retina II: Calculation of a spatio-temporal energy budget in retinal pigment epithelium and photoreceptor cells based on quantification of cellular processes

PONE-D-24-23593R1

Dear Dr. Kiel,

We’re pleased to inform you that your manuscript has been judged scientifically suitable for publication and will be formally accepted for publication once it meets all outstanding technical requirements.

When checking the correction print please address this: "Line 181: is the parameter “3 x 106 = (proteins/μm3)” intended to be on its own line?"

Kind regards,

Olaf Strauß

Academic Editor

PLOS ONE

Additional Editor Comments (optional):

Reviewers' comments:

Reviewer's Responses to Questions

**Comments to the Author**

1. If the authors have adequately addressed your comments raised in a previous round of review and you feel that this manuscript is now acceptable for publication, you may indicate that here to bypass the “Comments to the Author” section, enter your conflict of interest statement in the “Confidential to Editor” section, and submit your "Accept" recommendation.

Reviewer #2: (No Response)

Reviewer #3: All comments have been addressed

2. Is the manuscript technically sound, and do the data support the conclusions?

Reviewer #2: Yes

Reviewer #3: Yes

3. Has the statistical analysis been performed appropriately and rigorously? 

Reviewer #2: N/A

Reviewer #3: Yes

4. Have the authors made all data underlying the findings in their manuscript fully available?

Reviewer #2: Yes

Reviewer #3: Yes

5. Is the manuscript presented in an intelligible fashion and written in standard English?

Reviewer #2: Yes

Reviewer #3: Yes

6. Review Comments to the Author

Reviewer #2: The authors have addressed my comments to the extent they are able. The section they added describing limitations in the study was well done and relays to readers the fact that this paper is an important starting point when it comes to a more quantitative analysis of photoreceptor and RPE energy demands. One minor question / comment:

Line 181: is the parameter “3 x 106 = (proteins/μm3)” intended to be on its own line?

Reviewer #3: I have read the revised manuscript and the author responses. The authors addressed each of the concerns that I raised.

7. PLOS authors have the option to publish the peer review history of their article (what does this mean?). If published, this will include your full peer review and any attached files.

Reviewer #2: **Yes: **Daniel T. Hass

Reviewer #3: **Yes: **James Bryant Hurley

---

## [Editor Report · Acceptance letter]

19 Sep 2024

PONE-D-24-23593R1 

PLOS ONE

Dear Dr. Kiel, 

I'm pleased to inform you that your manuscript has been deemed suitable for publication in PLOS ONE. Congratulations! Your manuscript is now being handed over to our production team.

Kind regards, 

on behalf of

Professor Olaf Strauß 

Academic Editor

PLOS ONE